# A Signal Period Detection Algorithm Based on Morphological Self-Complementary Top-Hat Transform and AMDF

**Zhao Han and Xiaoli Wang \***

School of Mechanical, Electrical and Information Engineering, Shandong University, Weihai 264209, China; hanzhao@mail.sdu.edu.cn
\*  Correspondence: wxl@sdu.edu.cn; Tel.: +86-138-6302-6640

**Abstract:** Period detection technology for weak characteristic signals is very important in the fields of speech signal processing, mechanical engineering, etc. Average magnitude difference function (AMDF) is a widely used method to extract the period of periodic signal for its low computational complexity and high accuracy. However, this method has low detection accuracy when the background noise is strong. In order to improve this method, this paper proposes a new method of period detection of the signal with single period based on the morphological self-complementary Top-Hat (STH) transform and AMDF. Firstly, the signal is de-noised by the morphological self-complementary Top-Hat transform. Secondly, the average magnitude difference function of the noise reduction sequence is calculated, and the falling trend is suppressed. Finally, a calculating adaptive threshold is used to extract the peaks at the position equal to the period of periodic signal. The experimental results show that the accuracy of periodic extraction of AMDF after Top-Hat filtering is better than that of AMDF directly. In summary, the proposed method is reliable and stable for detecting the periodic signal with weak characteristics.

**Keywords:** weak characteristic signal; period detection; single period; self-complementary Top-Hat transform; average magnitude difference function; adaptive threshold

## 1. Introduction

The weak characteristics signal period detection in noise background has a very important position in many engineering applications, such as pitch detection in speech processing [1], vibration period of rolling bearing impact fault extraction in mechanical engineering [2], the vibration condition of motor stator and rotor fault detection in the power system [3], etc.

At present, there are many more classic period extraction algorithms such as the autocorrelation function (ACF) [4], average magnitude difference function (AMDF) [5–7], cepstrum [8], wavelet transform [9], etc. Among all period detection techniques, those based on AMDF and ACF are more widely used than other methods. AMDF has the advantages of low computation complexity and high precision, but with the increase of delay, the decrease of frame overlap for its calculating will lead to the phenomenon of a falling trend [10,11]. In addition, the traditional AMDF will appear to mistake showing the location of the lowest notch besides zero point rather than showing the real period. The result of the ACF method is similar with the AMDF's, which is the extreme points located at integer multiple of the detection signal period. However, the choice of these points will be interfered with by many factors, such as the size of signal frame, the type of window function and the reduction of the average. Therefore, the errors of double frequency and half frequency resulted from using the ACF will always appear in practical applications. For the problems that appeared in the above methods,

researchers have proposed some improved algorithms. Shimamura proposed weighted autocorrelation (WAC) that using AMDF weight ACF makes the peaks of pitch period more prominent in the speech pitch period extraction process [12]. But the results of that are sometimes inaccurate, for the falling trend of the mean value of AMDF will lead to an unreasonable weighting factor. The extended average magnitude difference function (EAMDF) proposed by Muhammad spreads over the second half of the previous frame, the current frame, and the first half of the next frame [13]. While that is a good correction of the falling trend in AMDF, it doesn't solve the problem of misjudgment caused by false peaks in period extraction fundamentally, so this method always generates errors in doing so. In contrast, this paper chooses AMDF with lower computational complexity as the main period detection method.

The AMDF method is susceptible to random noise [14,15], resulting in the notch not being obvious enough, so it needs a filtering method to denoise the signal in advance. Among the many filtering methods, mathematical morphology filtering is a kind of time-domain nonlinear filtering method with clear physical meaning, practicality and high efficiency, which is widely employed in some fields, such as power system signal processing [16], mechanical fault diagnosis [17–19], electrocardiogram (ECG) measurement [20] and image processing [21–23]. Self-complementation Top-Hat (STH) transform is a denoising method based on some operations of that, which is good at suppressing the background noise to a great extent and retaining the details of the original signal [24]. Therefore, in order to improve the shortcomings of traditional AMDF method, this paper proposes a period detection scheme applied to single period signals which combines AMDF with morphology self-complementation Top-Hat transform. In this paper, this method is applied to detect the period of weak pressure signal in the gas outlet of the mechanical diaphragm gas meter. This signal has periodicity due to the cyclical variation of the internal working state of the diaphragm gas meter. Also, when the gas flow rate is stable, its period is fixed. The results show that the method can extract the period accurately, and verify its accuracy and low computational complexity.

The rest of this paper is organized as follows. In Section 2, some related principles, and the proposed signal processing process combining AMDF with STH transform are presented. Section 3 applies the improved method to experiments. Section 4 analyzes the experimental results. Conclusions and remarks on possible further work are given finally in Section 5.

## 2. Related Theory and Improved Scheme

### 2.1. Principle of AMDF Algorithm

The conventional AMDF is defined as [25]:

$$D(\tau) = \sum_{n=0}^{N-\tau-1} |x(n) - x(n+\tau)|, \tag{1}$$

where $x(n)$ are the frames of input signal whose length is $N$, $\tau$ is the lag number.

$D(\tau)$ has the same characteristic of period with the original signal, and there are notches located in the position of $T_\mathrm{p}$ and its multiple where $T_\mathrm{p}$ is the period of the periodic or quasi periodic original signal. Therefore, the period of the useful signal can be determined by calculating $D(\tau)$ to find the position of the highest notch (expect zero position).

### 2.2. Mathematical Morphology Filtering

The principle of mathematical morphology is designing a structure operator which could move constantly in the signal and match the signal exactly, suppress the noise and keep the detail. The morphology transformation includes several kinds of basic operations, the eroding operation, dilating operation, open operation, close operation, and the combination of them [26]. This method was mainly used in image processing at the initial stage. In recent years, it has been used in one-dimensional signal filtering and has achieved good results [27–30]. Suppose that $x(n)(n = 0, 1, \dots, N-1)$ is

one-dimensional signal, $g(m)(m = 0, 1, \ldots, M - 1)$ is the structure operator, and $N$ is greater than $M$. Then the dilating operation $\oplus$ and the eroding operation $\ominus$ are expressed as [28]:

$$
\begin{aligned}
(x \oplus g)(n) &= \max\{x(n + m) - g(m)\}, \\
(x \ominus g)(n) &= \min\{x(n + m) - g(m)\}.
\end{aligned}
\tag{2}
$$

Based on the dilating operation and eroding operation, open operation $\circ$ and closed operation $\bullet$ are formed. The open operation can suppress and eliminate the positive impulse noise in the signal, while the closed operation can filter the negative impulse noise. They are expressed as [29]:

$$
\begin{aligned}
(x \circ g)(n) &= (x \ominus g \oplus g)(n), \\
(x \bullet g)(n) &= (x \oplus g \ominus g)(n).
\end{aligned}
\tag{3}
$$

### 2.3. Self-Complementation Top-Hat (STH) Transform

The morphology Top-Hat transformation includes the white Top-Hat (WTH) operation and the black Top-Hat (BTH) operation. WTH and BTH are expressed as:

$$
\begin{aligned}
\mathrm{WTH}(n) &= x(n) - (x \circ g)(n), \\
\mathrm{BTH}(n) &= (x \bullet g)(n) - x(n).
\end{aligned}
\tag{4}
$$

STH is defined as the sum of WTH and BTH, which is [29]:

$$
\mathrm{STH}(n) = \mathrm{WTH}(n) + \mathrm{BTH}(n) = (x \bullet g)(n) - (x \circ g)(n).
\tag{5}
$$

In one-dimensional signal processing, STH can increase the peak value of the signal, and then enhance the impact characteristics of the signal, which is conducive to peak extraction. In addition, the selection of the size and shape of structure operator has a great influence on the signal processing results in morphology operations. There are many kinds of structure operators with different shapes, such as triangles, circles, and other polygons and their combinations. These shapes should approximate the graphical characteristics of the detection signal, and the computation complexity should also be taken into account when making a selection [30].

### 2.4. An Improved Method Based on AMDF and STH

In order to solve the problems that the falling trend of AMDF is serious and the amplitude of AMDF is susceptible to noise, this paper combines the traditional AMDF method with the mathematical morphological filtering method STH to form a new periodic detection method, which can be called STH-AMDF. This method is mainly divided into three steps as follows.

(1)  Collect the periodic signal to be detected, and filter that by STH transform.
(2)  Use AMDF transform the filtered signal, and suppress the falling trend.
(3)  Set the dynamic threshold to extract peaks.

In step (1), a structure operator needs to be selected before morphology filtering. Since the noise in the collected signal is mainly from the rotating structure inside the diaphragm gas meter, a simple flat zero structure (linear structure) element is selected for filtering [17,24]. In step (2), to suppress the falling trend, this paper divides the AMDF value by the number of overlapping points. And then, the notches are converted to peaks by taking the reciprocal of the non-zero amplitude. This improved AMDF based on Formula (1) is defined as

$$
D(\tau) = \frac{\tau}{\displaystyle\sum_{n=0}^{N-\tau-1} |x(n) - x(n + \tau)|},
\tag{6}
$$

where all variables have the same meaning as Formula (1). $D(\tau)$ has the same characteristic of period with the original signal, and there are peaks located in the position of $T_\mathrm{p}$ and its multiple.

The periodic signal used in this paper is the pressure signal of the outlet of the traditional diaphragm gas meter. Since that is related to the model and structure of the gas meter, gas network pressure and background noise, the signal has no fixed shape, which also results in having no fixed range of AMDF spectrum values. Therefore, the step (3) uses the dynamic threshold method to extract the peaks in the AMDF spectrum. The flow chart of the peak extraction algorithm is shown in Figure 1.

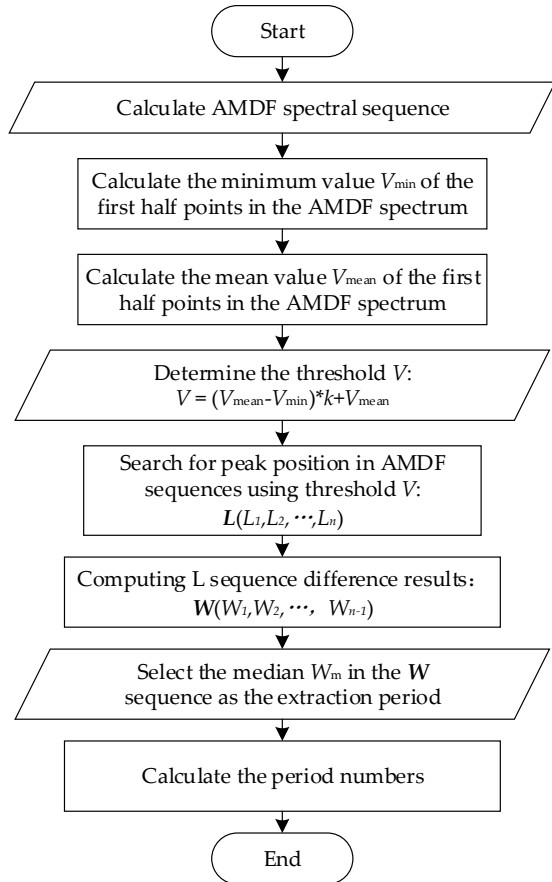

**Figure 1.** The flow chart of the peak extraction algorithm.

Figure 1 indicates that support for the length of AMDF spectral sequence is $N$. To extract the peak position and peak interval in that sequence, the minimum value $V_\mathrm{min}$ and mean value $V_\mathrm{mean}$ of the first to the $N/2$ points in the sequence are calculated first, and then calculate the peak extraction threshold $V$. The expression for $V$ is

$$V = (V_\mathrm{mean} - V_\mathrm{min}) \times k + V_\mathrm{mean}, \tag{7}$$

where $k$ is the coefficient of threshold calculating. The principle of Formula (7) is to screen out the peaks that are above a certain range of the mean of the spectral sequence, and the method of determining this range refers to the difference between the mean and the minimum in the spectrum.

Setting an appropriate threshold and performing peak identification and extraction of the AMDF sequence play an important role in the calculation of the signal period. If the threshold is too small, the aperiodic peak value in the AMDF sequence will be introduced, so that the calculated value of the spectral peak interval is too small, resulting in the computational flow higher than the actual value. If the threshold is too large, some effective peaks in the AMDF sequence will be omitted, so that the calculated value of the spectral peak interval is too large, resulting in a computational flow lower than the actual value. The effective peak appearing in the AMDF spectrum of the gas cycle pressure

cycle signal of the membrane gas meter is usually a narrow peak. The peak height is increased by reducing the signal noise, and the increase of the peak height does not have an excessive influence on the calculation of the $V_{mean}$ value. Therefore, the robustness of the signal cycle extraction algorithm can be improved by increasing the threshold coefficient *k* appropriately. Through a large number of calculation tests, the *k* of this paper is 1.4.

Next, the position sequence *L* is obtained by selecting the peak value, and the interval sequence *W* between the peaks is obtained by *L*. Then the median $W_m$ of the *W* sequence is selected as the period length of the signal. Finally, the number of periods is extracted by that length.

Summarizing the above steps, the period detection method proposed in this paper is shown in Figure 2.

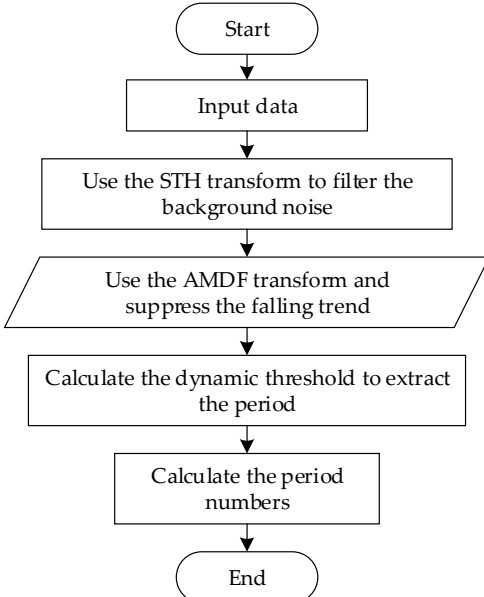

**Figure 2.** The flow chart of the peak detection method based on AMDF and STH algorithm.

## 3. Experimental Process and Results

In this paper, the measured periodic pressure signals are collected from the outlet of the mechanical diaphragm gas meter with the stable gas source, and the fluctuation of the pipeline network is eliminated. The sampling frequency is 6 times/second, and the known period range is 5 seconds to 90 s.

### 3.1. Simulation of Filtering with STH Transform

In the actual signal collecting process, there are many sources of noise and they are random. At the same time, for the collected signal, it's difficult to get the signal without noise. As such, the random noise is added to the collected signal, and then the morphological transform is used to filter the signal to verify its effect. Additive white Gaussian noise is a good way to simulate a variety of noise sources, so it's used in this simulation. In addition, the mean value of the collected signal is near 0, and the energy is low. The specific details of the experiment are as follows.

The signal with a length of 200 to be measured and the addition of Gaussian white noise (0, 2) are shown in Figure 3.

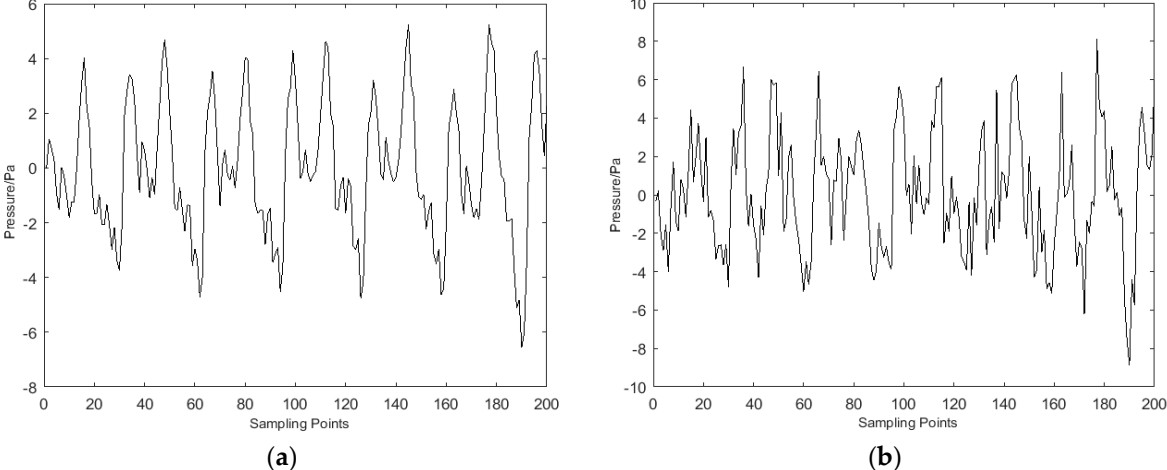

**Figure 3.** Pressure signal before filtering: (**a**) The original signal; (**b**) The signal after adding Gaussian white noise.

Figure 3 demonstrates that the original signal has periodicity and the peak value is obvious, but when the Gauss noise is added, the peak value is submerged in the noise. According to the reference [30], the length of the structural element is related to the period of the signal. Since the period length range is 30 points to 540 points, this paper selects a linear structural element with a length of 11 points to perform morphological filtering on the signal, which can remove the noise component contained in the signal and retain the characteristic pulse component. Firstly, the morphological open operation is performed on the signal with Gaussian white noise, and the peak in the original signal can be eliminated, and the fluctuation component with the width larger than the structural element is filtered out, and then the signal is compared with the calculation result of the open operation to extract positive peaks in the signal; the morphological close operation of the noisy signal can obtain the negative peaks, and finally the STH sequence containing the positive and negative peaks information of the original signal can be calculated, as shown in Figure 4.

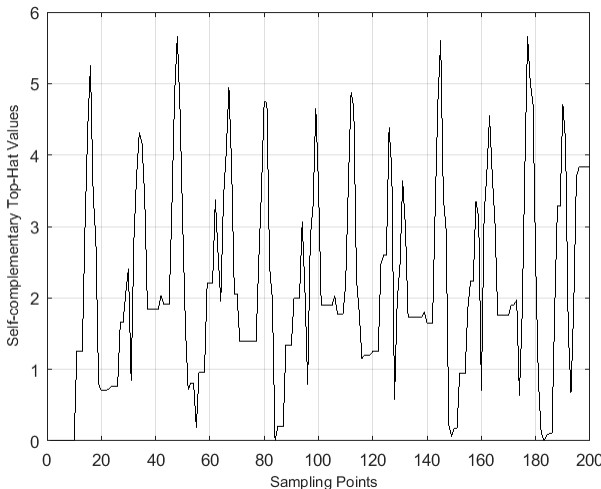

**Figure 4.** Results of self-complementary Top-Hat transform for the noisy signal.

It can be seen from Figure 4 that the peaks of the noisy signal after STH transformation become very obvious, which is very beneficial for the peak extraction.

The mathematical morphology filtering method is used to denoise the gas pressure signal, which has the characteristics of low computational complexity, good denoising characteristics and low-pass characteristics. Its core is to detect the position of the target in the signal by structural elements, obtain

the geometric shape information and its relationship in the signal, and then realize the judgment of signal characteristics. In addition, selecting the most commonly used flat structure elements as filtering windows can effectively extract useful features from the signal and further reduce the computational complexity of the algorithm.

### 3.2. Simulation of Improved AMDF and Peak Extraction Algorithm

A gas pressure signal segment with a length of 300 and its traditional AMDF calculation results are presented in Figure 5.

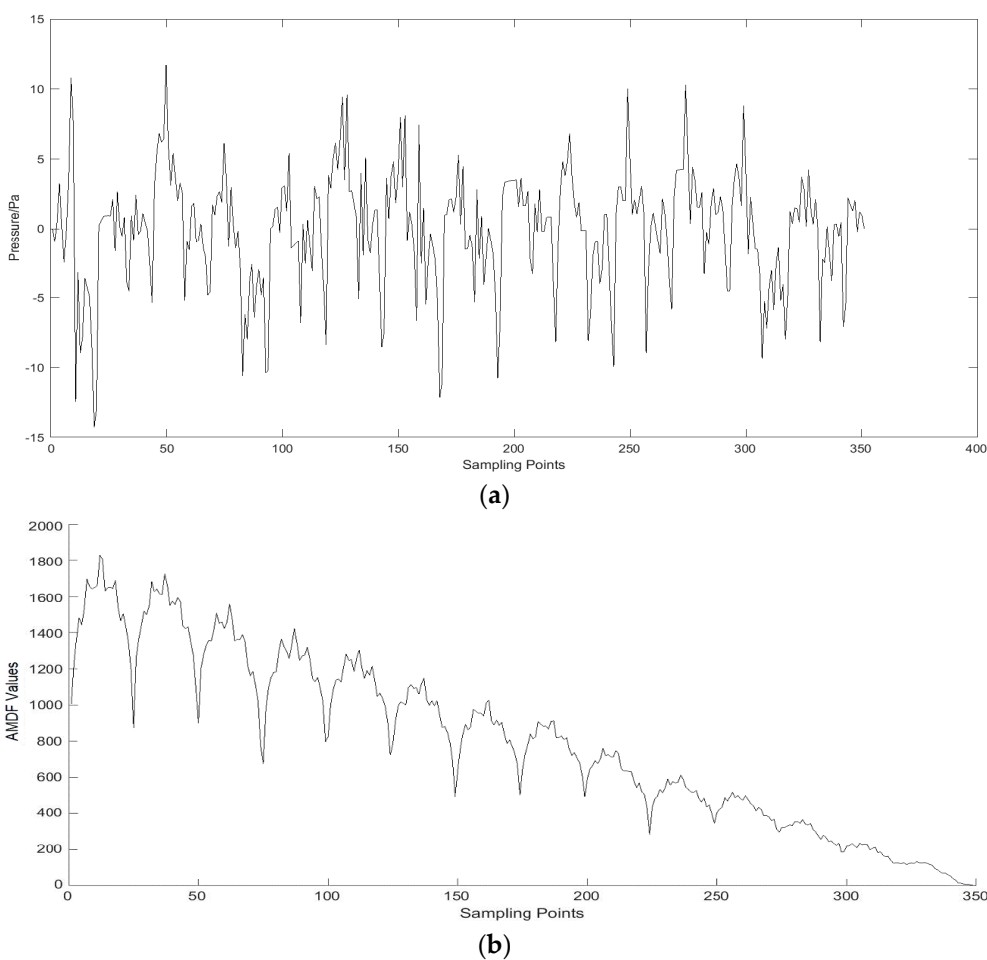

**Figure 5.** Simulation results of the traditional AMDF algorithm: (**a**) The original signal; (**b**) The traditional AMDF result.

According to Figure 5, with the increase of lag time, the peak amplitude of the conventional AMDF will decrease. Therefore, in this paper, the improved AMDF method (Formula (6)) is applied, and the result of that is shown in Figure 6.

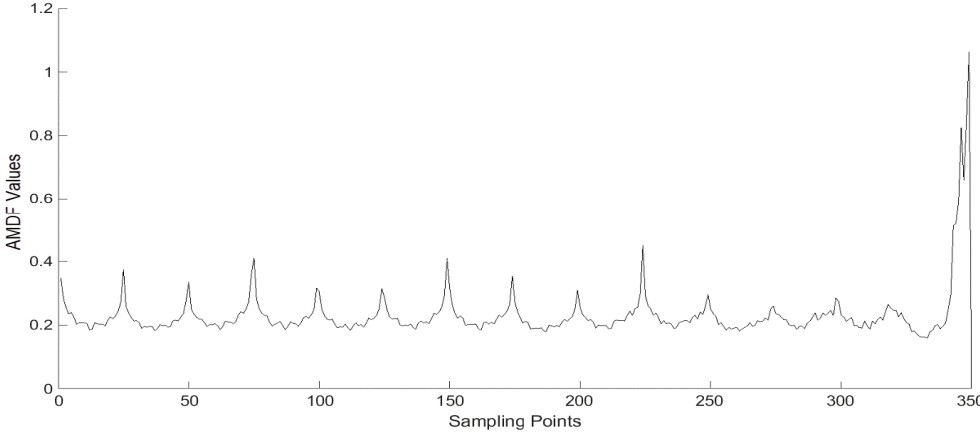

**Figure 6.** Simulation results of the improved AMDF algorithm.

Figure 6 indicates that the improved AMDF method can make the periodic characteristic of the signal be expressed in the form of peaks and eliminate the downward trend of the traditional AMDF calculation results. In addition, as can be seen from Figure 6, peak values are different and it is difficult to extract the peaks by using fixed threshold, so the dynamic threshold is calculated by using the algorithm shown in Figure 1.

Figure 7 is the simulation result diagram of the AMDF peaks extracting method. The red line is the threshold line. It can be seen from the figure that this method is effective in extracting the peaks. However, the peak amplitudes are still not obvious, so the determined threshold cannot extract all peaks.

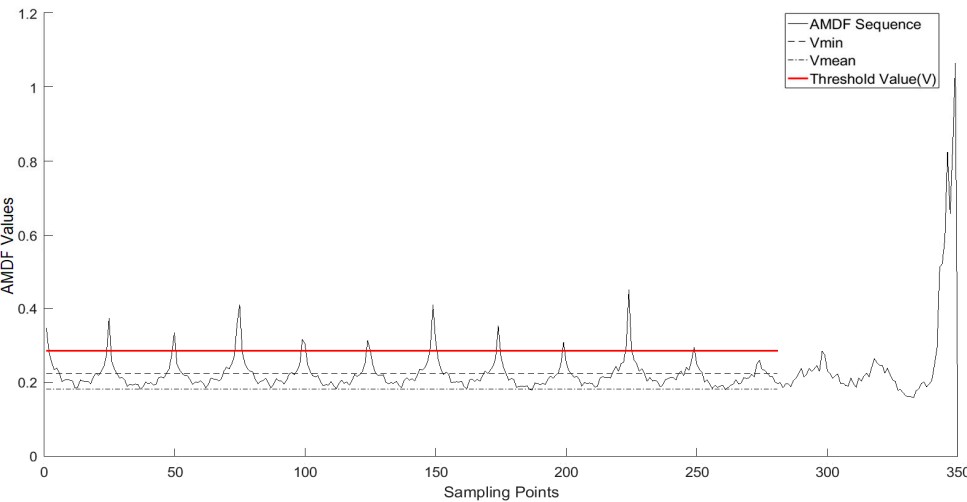

**Figure 7.** Simulation results of the peak extraction algorithm.

### 3.3. Simulation of STH-AMDF Algorithm

In order to make the experimental results more obvious, this paper selects a sampling signal of length 700 (Figure 8a). In the first group, AMDF operation and peak extraction were performed directly on the sampled signal (Figure 8b). In the second group, STH filtering was performed on the sampled signal (Figure 8c), followed by AMDF operation and peak extraction (Figure 8d).

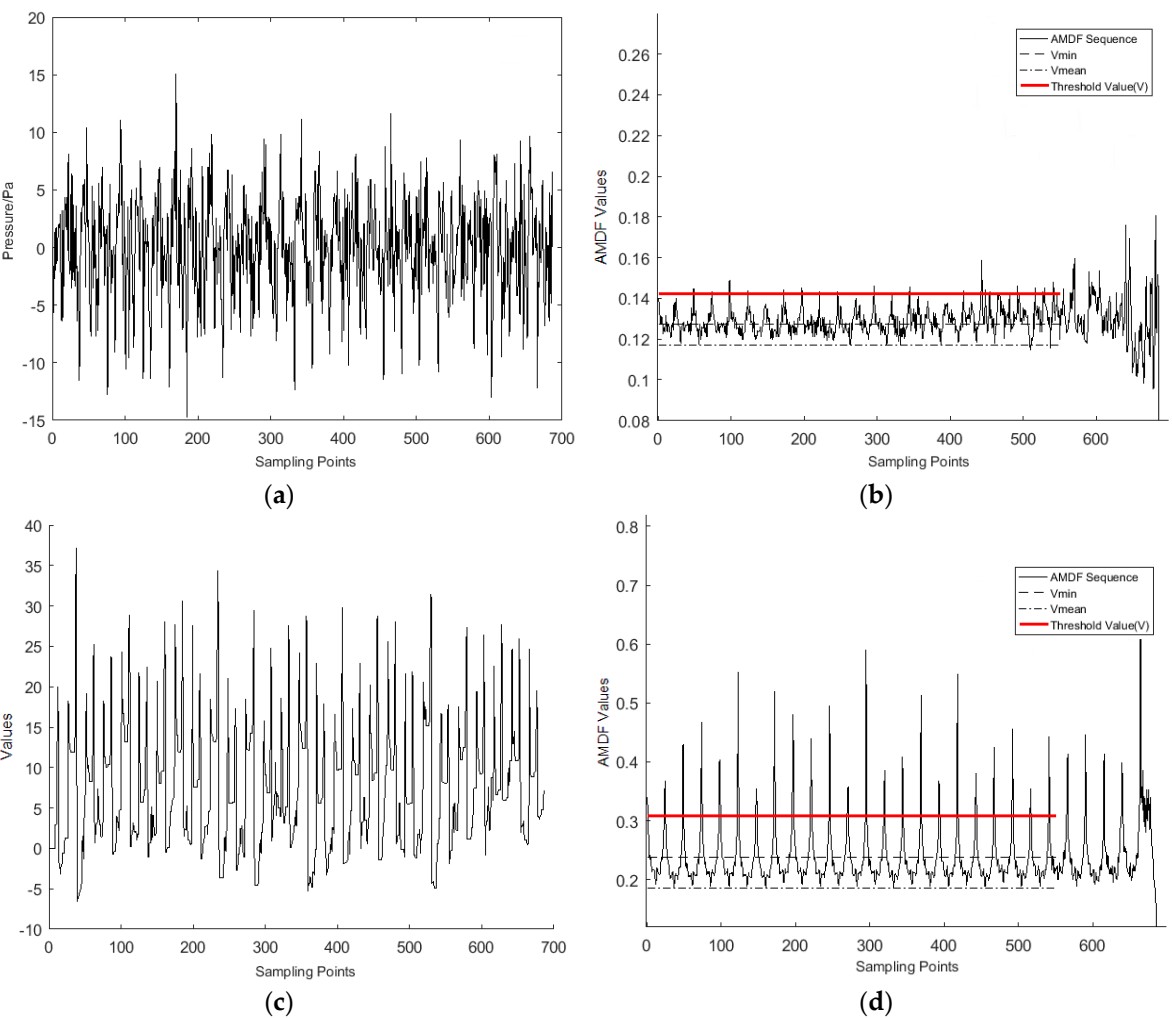

**Figure 8.** Simulation results of the STH-AMDF algorithm: (**a**) The original signal; (**b**) The result of AMDF of original signal and peak extraction algorithm; (**c**) The signal after STH transform; (**d**) The result of STH-AMDF algorithm.

As shown in Figure 8b, the second, sixth, eleventh, fifteenth and sixteenth peaks in the AMDF spectrum are lower than the extraction threshold, and the peaks after the nineteenth peak are pseudo-peaks and their peak intervals are significantly smaller than the real periodic values. Although the error of this calculation can be avoided by using the median of the peak interval as the period value, when the signal data is shortened, the probability of occurrence of the cycle calculation error will increase greatly. Therefore, the signal needs morphological filtering to enhance the useful peak information of the original weak periodic signal and filter out part of the noise. According to Figure 8d, after morphological filtering, the AMDF spectral shape of the signal is clearly improved for extracting periodic features, which is manifested by the distinct increase of periodic peaks, uniform distribution, reduction of pseudo-peaks and low height. The threshold set by Formula (7) is approximately in the middle of the effective peak, higher than the pseudo-peaks and lower than all the effective peaks. This method greatly improves the accuracy and robustness of the weak signal periodic extraction algorithm.

## 4. Discussion

In order to verify the optimization effect of self-complementary Top-Hat filter on AMDF peak extraction, it is necessary to calculate the mean of effective peak height $m_p$ in AMDF spectrum and the

mean value $m_\mathrm{n}$ in spectrum sequence with the effective peak removed. The definition of the peaks recognition accuracy parameter $P$ is as follows:

$$P = \frac{m_\mathrm{p}}{m_\mathrm{n}}. \tag{8}$$

The larger the $P$ value is, the higher the pseudo-peak value is, and the easier it is to filter the pseudo-peak and reduce the pseudo-peak interference. The smaller the $P$ value is, the closer the effective peak height is to the pseudo peak, which increases the probability of periodic error extraction. When the $P$ value is small, it usually leads to the extraction period value being smaller than the actual value, which leads to the flow calculation value being larger than the actual value. The experimental data were collected from the pressure signals of the outlet of five different specifications of household diaphragm gas meters under different fire conditions. Each test group contains five complete signal cycles. The accuracy parameter $P$ from using the AMDF method and the STH-AMDF method is shown in Figure 9.

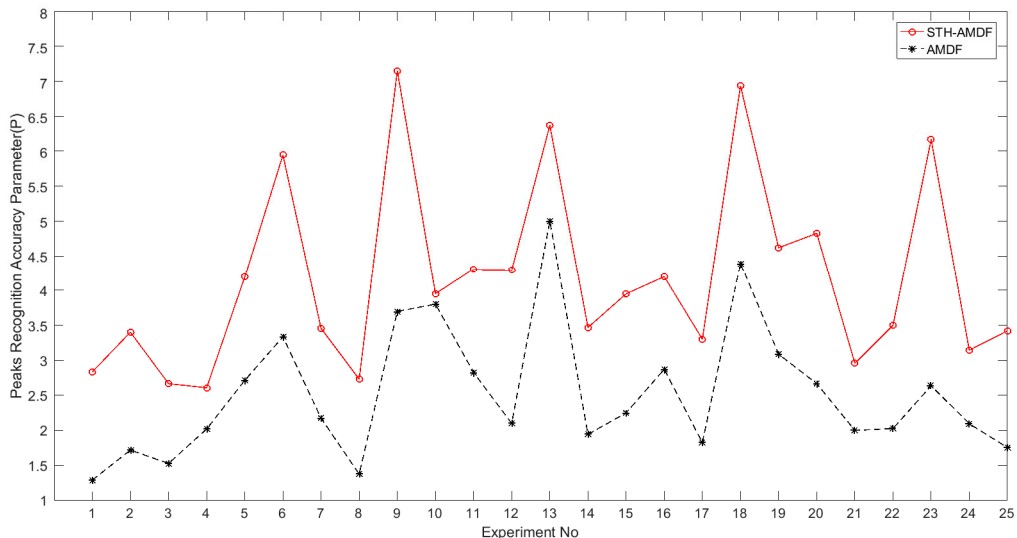

**Figure 9.** The peaks recognition accuracy parameter distribution using the AMDF and STH-AMDF methods.

The Figure 9 shows that the effective peaks in STH-AMDF sequences with morphological filtering are easier to extract than those in AMDF sequences. The STH-AMDF period extraction algorithm can better avoid the calculation errors caused by the interference peaks, and it is convenient to set a reasonable extraction threshold to ensure the accurate identification of the periodic peaks, so as to achieve accurate detection of the signal period.

## 5. Conclusions and Future Work

In order to extract the periodic number of noisy weak characteristic periodic signals more accurately, a STH-AMDF method of extracting period based on morphological self-complementary Top-Hat transform and average magnitude difference function is proposed in this paper. Traditional AMDF can reflect the periodicity of the signal, but when there is noise, its peaks are not obvious enough, which is not conducive to peak extraction. Therefore, STH is applied to filter the noise and improve the impact characteristics of the signal. Experiments show that the improved algorithm can increase the peak amplitude and set dynamic threshold to extract peak value, which verifies the accuracy and effectiveness of this method.

In future work, we will do some significant researches on the following aspects. First, we will make a meaningful study about the real-time period detection for the signal of changing period.

Secondly, we will collect more pressure data of the exhaust port of domestic gas meters and optimize the algorithm to make it suitable for family life.

**Author Contributions:** Z.H. proposed innovative idea; Z.H. and X.W. conceived the algorithm and wrote the first draft; X.W. improved the algorithm; Z.H. performed the experiments; Z.H. and X.W. analyzed the results; Z.H. drafted the manuscript; X.W. provided writing advice; Z.H. and X.W. read and approved the final manuscript.

**Funding:** This research received no external funding.

**Acknowledgments:** We gratefully acknowledge the technical assistance of DL850E ScopeCorder.

**Conflicts of Interest:** The authors declare no conflict of interest.

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
