# Peer review of "A Signal Period Detection Algorithm Based on Morphological Self-Complementary Top-Hat Transform and AMDF"

_information, doi:10.3390/info10010024_

Round 1

Reviewer 1 Report

First of all let me thank the authors for driving attention to such a basic but extremely relevant problem in signal processing. I appreciate their work and desire to contribute to such a relevant, but at the same time well-known problem.

That being said, I must strongly encourage the authors to send their manuscript for review. The written has numerour flaws, odd expressions and incomplete phrases (for instance, in the abstract, authors write "However, the effect of this method is not very well when the...". Don't really get "very well what"). Until writing is improved, the manuscript won't be ready for publication.

For the content of the manuscript, I have several major and a few minor comments.

Major concerns:

- signal period detection being a well-known problem, is currently a relatively broad topic. For instance, single or multiple periods (as well as pitchs) may be computed, considering different applications. I believe authors wnat to focus on single period (monophonic) applications, but this should be stated directly in the abstract & introduction sections.

With this idea in mind, please explain whether the application addressed ("weak pressure signal in the gas outlet of the mechanical diaphragm gas meter") shall display heavy periodicity (despite background noise), or not. And whether this is a single-period (monophonic) or multi-period (poliphonic) signal. 

- noise background is an extremely broad concept, most certainly conditioned by the application. Just considering speech applications, background noise will largely differ when addressing a noisy analog channel, or the noise in a street or a bar. In each of these applications, the definition for a "strong noise" may largely vary. Please provide examples and some number in the introduction, to ensure that readers approaching the topic may have a 360º view on the problem.

Regarding the particular application being addressed, please provide further information regarding the nature and magnitude of the noise components being faced. 

- authors are very keen in remarking that speed in period detection is as relevant for them as accuracy. Regarding the introduction section, please explain the need for this, and what their goal is concearing speed: real-time? 1.5 time real-time?

- the procedure describes in Section 2.4 states as part of step 1 "filter that by STH transform". Shouldn't be find a suitable structure operator prior to filtering? I couldn't find any reference to g (structure operator) in Section 2.4. Furthermore, (6) includes a parameter N, the value of which was not indicated. One could not follow the procedure described without these. If this shall be adapted to the particular application at hand, please state so. 

- Please provide further discussion on the improved AMDF based on formula 1, and compare it to other (similar) approaches, such as the extended-AMDF. 

- Attending to equation (7), threshold selection appears to be linked to the statistics of the clean (noiseless) signal. In particular, when using a minimum value estimator one may like to further describe the underlying signal required / recommended statistics. 

- please explain the rationale behind the simulation in Section 3.1. Why should we consider a additive, white, gaussian noise? Why did author choose a zero-mean noise? What it the expected SNR or peak-to-noise ratio that explains the choice for noise power?

- Based on the discussion in lines 158-161 it may appear that the choice for the structure operator (shape and length) is "art" rather than science. Please provide further details so as to ensure that readers my adapt this method to their particular application. Furthermore, it appears that authors choice for the structura element length is largely linked to the actual period length. If so, this would be a serious drawback for the proposed method. Please provide further details. 

- Considering the connection of the problem to the underlying, background noise, authors may like to extend the simulation in Section 3.2 with different noise levels. Both to prove the value of their work, and to evidence the point at which the proposed procedure no longer provides reliable estimations.

- Limited information is provided on the simulations included in Section 3.3, regarding the underlying signal or the noise that was used. Additionally, please extend the simulation in the same way as for Section 3.2.

- In addition to the discussion of the results provided in lines 213-220, it would be much appreciated that authors included results while varying the parameters of their algorithm.

- In the same, comparison of alternative algorithms (not just standard AMDF), would enhance the value of their contribution.

Minor concerns:

- In section 2.1, authors write "f is the original signal frequency without noise". Then this must be a pure sinusoid? Please make sure that the frequency related to periodicity is not mistaken with the different frequencies where energy (spectral content) may be allocated. 

- Regarding Mathematical Morfology Filtering (section 2.2), please include further references to evidence the use of this kind of processing on 1D signals. This has certainly become very common in recent years, as authors claim.

- It is not fairly standard to use greek letter Theta as an operator. I believe authors are looking for character CIRCLED MINUS' (U+2296). Please change this to improve readability. 

- open and close operations are pretty standard image processing applications, but not so widely used on 1D signals. To ensure that novel readers may have a feeling on how these operators are expected to behave, please include some examples in Section 2.2.

- In figures 5, 6 and 7, I believe y-axis should read "values", not "valus".

Author Response

Dear Professor,

Thanks for your comments. Our reply to your comments is attached. Thank you very much.

Best regards.

Your sincerely,

Zhao Han and Xiaoli Wang*

Reviewer 2 Report

This is a well presented study concerning a general problem of noise removal for many applications. Therefore, I believe, it should be of interest to a wide group of readers. The proposed method is clear, although the novelty is average - it can be concluded based rather easily based on the current state of the art. However, the presentation, simulations and conclusions are good. Therefore, I recommend publication. 

Author Response

Dear Professor,

Firstly, we would like to thanks the reviewer for the positive and constructive comments. According to your comments, we have checked our manuscript carefully. Some grammatical & language errors and other inexact expressions in the manuscript have been corrected. The important changes in our revised manuscript have been marked “in Red”. Thanks again.

Best regards.

Your sincerely,

Zhao Han and Xiaoli Wang*